# Effect of Physical Activity on Metabolic Syndrome Markers in Adults with Type 2 Diabetes: A Systematic Review and Meta-Analysis

**DOI:** 10.3390/sports11050101

**Published:** 2023-05-09

**Authors:** Mohammed Amin, Debra Kerr, Yacoba Atiase, Rami Kamel Aldwikat, Andrea Driscoll

**Affiliations:** 1Centre for Quality and Patient Safety, Institute for Health Transformation, School of Nursing and Midwifery, Faculty of Health, Deakin University, 221 Burwood Highway, Burwood, VIC 3125, Australia; d.kerr@deakin.edu.au (D.K.); rami.aldwikat@monash.edu (R.K.A.); andrea.driscoll@deakin.edu.au (A.D.); 2National Diabetes Management and Research Centre, Korle-Bu Teaching Hospital, University of Ghana Medical School, Accra P.O. Box GP 4236, Ghana; yatiase@ug.edu.gh; 3School of Nursing and Midwifery, Monash University, 35 Rainforest Walk, Clayton, VIC 3800, Australia

**Keywords:** physical activity, exercise, type 2 diabetes, systematic review, meta-analysis, metabolic syndrome, cardiovascular risk factors, adults

## Abstract

People with Type 2 diabetes mellitus (T2DM) are reported to have a high prevalence of metabolic syndrome (MetS), which increases their risk of cardiovascular events. Our aim was to determine the effect of physical activity (PA) on metabolic syndrome markers in people with T2DM. The study design was a systematic review and meta-analysis of randomised controlled trials evaluating the effect of PA on MetS in adults with T2DM. Relevant databases including SPORTdiscus, Cochrane Central Register of Controlled Trials, CINAHL, MEDLINE, PsycINFO, EMBASE, SocINDEX were searched up to August 2022. Primary endpoints were changes in MetS markers (blood pressure, triglyceride, high-density lipoprotein, fasting blood sugar, and waist circumference) after an exercise intervention. Using a random effect model with 95% confidence interval (CI), the mean difference between intervention groups and control groups were calculated. Twenty-six articles were included in the review. Overall, aerobic exercise had a significant effect on waist circumference (Mean Difference: −0.34 cm, 95% CI: −0.84, −0.05; effect size: 2.29, *I*^2^ = 10.78%). The effect sizes on blood pressure, triglyceride, high-density lipoprotein, fasting blood sugar were not statistically significant. No significant differences were found between exercise and control group following resistance training. Our findings suggest that aerobic exercise can improve waist circumference in people with T2DM and MetS. However, both aerobic and resistance exercise produced no significant difference in the remaining MetS markers. Larger and higher-quality studies are required to determine the full effects of PA on MetS markers in this population.

## 1. Introduction

Globally, about 425 million people are living with diabetes, and it is estimated that this figure will increase to about 629 million by 2045 [1]. Diabetes-related death is also estimated to double by the year 2030 [2]. Type 2 diabetes mellitus (T2DM) constitutes more than 90% of diabetes cases and is associated with high morbidity and mortality [3,4]. Metabolic syndrome (MetS) is a complication of poorly managed T2DM [5], although the syndrome can also act as a precursor to T2DM [6]. MetS is a collection of cardiovascular risk factors that occur together and increase an individual’s risk for cardiovascular disease (CVD), stroke, and kidney failure [7]. 

People with T2DM have an increased risk of developing MetS compared to individuals without T2DM [8]. MetS is associated with high mortality in people with T2DM [9]. However, the diagnosis of MetS is difficult due to varying definitions and diagnostic threshold values. According to the National Cholesterol Education Program Adult Treatment Panel (NCEP-ATP) [10] III definition, MetS can be confirmed if an individual has at least three of the following five conditions: high blood pressure (BP), high triglycerides, reduced high-density lipoprotein (HDL) levels, high fasting blood glucose (FBG) level, and increased waist circumference (WC). Although each component independently increases the risk of developing CVD, the combination of these factors significantly raises this risk even more [8].

There is evidence of increasing rates of MetS among T2DM clients [11]. A cross-sectional study in Syria among 424 diabetes patients aged between 40 and 79 revealed a MetS prevalence rate of about 68%, with females recording a higher prevalence than males [12]. A study among diabetes patients in Palestine showed a relatively higher prevalence rate (88.2%) of MetS compared to the study in the Syrian population [13]. A cross-sectional study among people with diabetes in China found that 66.7% of the study population had MetS [5]. In Ghana, a cross-sectional study involving 405 people with diabetes at the Komfo Anokye Teaching Hospital found that 90.6% had MetS [14].

Intervention programs aimed at controlling MetS in people with T2DM is an urgent priority [15]. Physical activity (PA) has been identified as an integral component of T2DM management [16]. Various exercise types (e.g., aerobics and resistance training) can be useful in achieving positive health outcomes including reduced episodes of hyperglycaemia, insulin sensitivity and improved BMI [17]. However, the effectiveness of PA interventions on MetS markers has not been well documented in people with T2DM, with varied study results [18]. There is the need to evaluate the effectiveness of PA interventions for reducing MetS markers in individuals with T2DM. For this systematic review and meta-analysis, we evaluated the effect of physical activity on metabolic syndrome markers (blood pressure, triglyceride, high-density lipoprotein, fasting blood glucose, and waist circumference) in people with type 2 diabetes.

## 2. Materials and Methods

The study protocol was registered with PROSPERO (CRD42020193566). The reported guideline adopted was the Preferred Reporting Items for Systematic Review and Meta-analysis (PRISMA) [19]. The review was guided by the Cochrane Handbook for Systematic Reviews of Interventions [20].

### 2.1. Types of Studies Included

The review was restricted to RCTs that explored the use of PA to improve MetS for individuals with T2DM. Studies included in the review were conducted in the community or clinical/facility settings.

Studies were included if they met the following criteria: (a) participants had a diagnosis of T2DM and at least 18 years of age, (b) measurement for MetS with specific evaluation of at least three of the following health outcomes: systolic blood pressure (SBP), diastolic blood pressure (DBP), triglyceride, HDL, FBG, and WC, (c) participants were followed up for at least 3 months, and (d) a comparison group was randomised to their usual diabetes treatment. 

Papers were excluded if they met the following criteria: (a) studies with a high attrition rate (over 20%) if the intention to treat analysis was absent, (b) studies with diet or pharmacological therapy as a co-intervention without independently assessing PA effect, and (c) studies that utilised PA advice or counselling only interventions. 

Intervention: For this study, PA was defined as any bodily movement produced by skeletal muscles that require energy expenditure higher than resting (Persson, 2015). Physical activity methods considered in the search were aerobic training (e.g., walking, jogging, running, cycling), resistance training (e.g., weightlifting, push-ups), and alternate exercise programs (e.g., pilates, yoga, tai chi), with varying degrees of intensity (moderate, vigorous), frequency and duration. Physical activity could be domestic-related (e.g., being active while performing household duties), work-related (e.g., being active during your job), transport-related (e.g., getting active whilst moving from one place to another) and planned exercise (e.g., aerobics, resistance training, yoga, pilates) [21]. Exercise programs may also have involved supervised or recommended home-based physical activities. 

Comparator: A control group in which participants followed their usual diabetes care.

Outcomes: The primary outcome was an improvement in any of the following MetS markers: SBP, DBP, triglyceride, HDL, FBS, and WC. 

### 2.2. Data Sources and Search Strategy

Relevant databases were searched for RCTs that included a PA intervention aimed at modifying MetS markers in adults with T2DM. A search strategy (Appendix A) was used to identify literature from three sources. Databases (SPORTdiscus, Cochrane Central Register of Controlled Trials (CENTRAL), CINAHL, MEDLINE, PsycINFO, EMBASE, SocINDEX) were searched up to August 2022. Reference lists of identified articles were hand-searched to identify other potential and relevant studies. Third, relevant literature was explored using trial registries such as the World Health Organisation International Clinical Trials Registry Platform, Cardiosource Registry of Randomised Cardiovascular Clinical Trials and The International Standard Randomised Controlled Trial Number Registry (ISRCTN). The search was limited to RCTs and published in English, and no date restriction was applied. 

### 2.3. Data Selection

One investigator (MA) conducted an initial search after which all duplicates were removed using ENDNOTE X9 [22] and DeDuplicator (Systematic Review Accelerator, Institute for Evidence-based Healthcare, Bond Univeristy, available at www.sr-accelerator.com, accessed on 20 March 2022). Then, two investigators (MA and RKMA) independently screened titles and abstracts to exclude studies that did not meet eligibility criteria. Full text of all papers was subsequently examined independently by two investigators (MA and RKMA) for eligibility. Disagreement regarding eligibility were discussed with the third investigator (DK) who adjudicated the matter. 

### 2.4. Data Extraction

Two investigators (MA and DK) independently extracted data from eligible papers using web-based software platform that facilitates systematic reviews (Covidence systematic review software, Veritas Health Innovation, Melbourne, Australia, available at www.covidence.org, accessed on 28 March 2022). A third investigator (AD) adjudicated any disagreement. Data were extracted using the Cochrane Handbook for Systematic Reviews of Interventions as a guideline [20]. Data extracted included study details (aim, design, setting, country, author (s), publication date), sample characteristics (population, sample size), MetS makers (DBP, SBP, WC, triglycerides, HDL and FBS), and PA description (type/duration/frequency). Data were extracted regarding the effect of PA on primary outcomes at baseline and follow-up. Data were then transformed from multiple data formats into a single data format to remove unwanted data including duplicates.

### 2.5. Risk of Bias Assessment

Selected studies were assessed for risk of bias using the Physiotherapy Evidence Database (PEDro) scale. The scale is used in evaluating the quality of RCTs assessing PA interventions, and the reliability of the total PEDro score is graded ‘fair’ to ‘good’ [23]. Study bias was assessed for eleven areas: specification of eligibility criteria, random allocation, allocation concealment, group similarity at baseline, blinding of subjects, therapist blinding, assessor blinding, point estimate reporting, dropout rate less than 15%, intention to treat analysis report, and reporting group difference. Each specified item on the scale contributes 1 point to the total PEDro score, and total scores range between 0 and11. A score less than 4 indicates a lower quality RCT [23]. 

### 2.6. Data Synthesis Strategy

Data were analysed using STATA (version 17, STATA Corp LLC, College Station, TX, USA). Descriptive statistics (mean, SD) were used to describe the study population, interventions, and outcome measures. A random effect model was employed for the meta-analysis in which mean difference of outcome measures were compared for intervention and control groups. Forest plots (Appendix A) were used to graphically present the results. The Hedge’s G was used to measure the effect size, with a *p*-value less <0.05 indicating a statistically significant difference between the control and intervention group. An effect size of 0.2, 0.5, and 0.8 were interpreted as small, medium, and large, respectively. Heterogeneity was measured using the *I*^2^ statistic. Heterogeneity is considered low, medium, or high when *I*^2^ percentages are around 25%, 50%, or 75%, respectively [24]. Funnel plots and Egger’s tests (where applicable) were used to test for publication bias.

## 3. Results

The PRISMA flow chart is shown in Figure 1. A total of 2652 studies were identified from initial database and hand searches. Then, 838 duplicates were removed. The remaining 1814 studies were imported into COVIDENCE for screening. Another 222 duplicates were removed. Out of the remaining 1592 studies, title and abstract screening identified 1388 irrelevant studies. A further 152 studies were excluded after full text screening. Papers were excluded for not meeting the inclusion criteria based on intervention (63 studies), outcome (76 studies), patient population (11 studies), comparator (8 studies), study design (16 studies), and duplicates (4 studies). Finally, 26 studies were included in the review.

### 3.1. Interventions

Details of included studies are shown in Appendix A. Three main forms of exercise interventions were trialled: aerobics, resistance, and combined (aerobics plus resistance). A total of 3336 participants were recruited in the studies. Samples ranged from 20 [25] to 794 participants [26]. 

Most PA programs (n = 17) were delivered in clinical settings or exercise laboratories. Seven programs were home/community based [26,27,28,29,30,31,32], and two were a combination of facility-based and home settings [32,33]. Only one study each was conducted in Africa [34] and Australia [30], and three studies were conducted in Europe [35,36,37]. Nine studies were conducted in Asia [25,26,29,38,39,40,41,42,43], five studies in Central/South America [33,44,45,46,47], and seven studies in North America [27,28,31,32,48,49,50].

Exercise duration ranged from 12 weeks [37,39,41] to 52 weeks [36,38]. The average exercise duration was 23 weeks.

Most PA interventions (n = 22) included a warm-up, main exercise, and cool-down. Moderate aerobic exercise included walking, jogging, stationary biking, stepping, running and aerobic dance [26,28,29,33,37,39,45,46,48]. Other aerobic exercises involved the use of equipment such as elliptical machines, treadmill, and cycle ergometer [31,44]. Low intensity aerobic exercise was achieved using stretching and flexibility exercises [32,47,50]. Resistance exercises included exercises such as chest press, lateral pull down, leg press, trunk flexion, biceps free weights, seated row, triceps, and flat bench press, legs extension, calf, seated leg curl, chair stands [27,36,37,42,43,47,49,50]. Combined training consisted aerobic plus resistance exercises [37,49,51].

### 3.2. Method Quality

Risk of bias for the included studies are summarised in Table 1. Based on the PEDro scale, no study achieved an excellent quality rating, most likely due to the difficulty in blinding participants for PA interventions. All studies met the following four criterion: specifying eligibility criteria, random allocation, point estimate reporting, and group difference reporting. No study had a total PEDro score less than four; one study had a score of four [38] and 23 studies had a score between five and eight. Two studies had a score of nine [42,49], indicating high methodological quality. Four studies included blinding to the therapist [30,41,42,49] and only one study blinded participants to the intervention [49]. Twelve out of the 26 studies achieved assessor blinding [25,27,30,33,36,41,42,43,47,49,50,52].

### 3.3. Meta-Analysis

Ten studies were included in the meta-analysis. Five studies utilised aerobic exercise, four studies used resistance exercise, and one study used both aerobic and resistance exercise. A total of 406 participants were recruited in those studies. Samples ranged from 23 [25] to 75 participants [26].

#### 3.3.1. Effect of Aerobic Training on Metabolic Syndrome Markers

Figure 2a–d and Appendix A show the mean changes for MetS markers (DBP, SBP, WC, HDL, triglycerides, and FBS), comparing the intervention group (IG) to the control group (CG). The mean changes were −0.55 mmHg (95% CI −1.53, 0.43; effect size: 1.09) for SBP and −0.61 mmHg (95% CI −1.75, 0.54; effect size: 1.04) for DBP: mean differences were not statistically significant. High heterogeneity was found for both SBP and DBP (DBP: *I*^2^ = 87.39%; SBP *I*^2^ = 90.41%). Regarding FBS, the IG recorded a change of −0.23 mg/dL (95% CI −0.55, 0.09; effect size: 1.44) compared to the CG, but the difference is non-significant. Low heterogeneity (*I*^2^ = 27.13%) was found in these studies. For triglycerides, the mean changes were −1.08 mmol/L (95% CI −2.85, 0.68, effect size: 1.20), with high heterogeneity (*I*^2^ = 96.5%) as shown in Appendix A. The mean change for HDL was 2.00 mmol/L (95% CI −1.40, 5.40; effect size: 1.15), with high heterogeneity (*I*^2^ = 99.2%) in those studies. This shows that the mean changes in triglycerides and HDL were statistically non-significant.

There was a statistically significant change in WC (MD −0.34 cm, 95% CI −0.84, −0.05; effect size: 2.29), with low heterogeneity (*I*^2^ = 10.78%). Thus, among the MetS markers, only WC was significantly improved through aerobic exercise.

#### 3.3.2. Effect of Resistance Training on Metabolic Syndrome Markers

Evidence of the effect of resistance training on various MetS markers (DBP, SBP, WC, HDL, triglycerides, and FBS), comparing IG to CG is shown in Appendix A. The mean changes were −0.06 mmHg (95% CI −0.34, 0.21; effect size: 0.45) for DBP and −0.43 mmHg (95% CI −0.87, 0.02; effect size: 1.87) for SBP. This indicates a trend towards reduction in both DBP and SBP in the IG compared to the CG. However, these differences were not statistically significant. Additionally, the analysis shows medium heterogeneity for SBP (*I*^2^ = 58.95%) and no heterogeneity for DBP (*I*^2^ = 0.0%). Regarding FBS, the IG recorded a change of −0.16 mg/dL (95% CI −0.43, 0.12; effect size: 1.10) compared to the CG, but the difference is non-significant. The analysis shows no (*I*^2^ = 0.0%) heterogeneity in those studies. Further, the mean changes were −1.03 mg/dL (95% CI −2.43, 0.38; effect size: 1.43) for triglycerides, 0.25 mg/dL (95% CI −0.03, 0.53; effect size: 1.78) for HDL and −0.24 cm (95% CI −0.52, −0,04; effect size: 1.71) for WC, with high heterogeneity (*I*^2^ = 92.78%) for triglycerides and no heterogeneity (*I*^2^ = 0.0%) for both HDL and WC.

## 4. Discussion

This review consisted of 26 RCTs involving a total of 3300 participants diagnosed with T2DM. Our meta-analysis showed that aerobic exercise interventions affect WC. However, no statistically significant change was observed for other MetS markers for aerobic or resistance exercise interventions.

Our finding that aerobics exercise improves WC in adults with T2DM is comparable to a study among healthy adults with MetS in which aerobic training led to significant improvements in WC (−3.4 cm) [53]. Though our current study showed a relatively small difference of means (2.29 cm), there was a significant improvement in WC. Further evidence has suggested that a reduction in WC, can potentially improve HDL compared to dietary interventions [54]. This is achieved via two mechanisms: (a) lipoprotein lipase facilitating transfer of lipid to HDL, and (b) hepatic triglyceride lipase reducing HDL clearance [54]. This means that a reduction in WC can potentially improve MetS.

Improvement in body composition has been found to correlate with positive changes for other MetS markers including FBG, Hb1Ac, SBP, and DBP [55,56]. For example, an improvement in body composition by 1 kg results in overall reduction in blood pressure by 1 mmHg [57] and 1.54 mg/dL in triglycerides [56]. Improvement in WC also has the potential benefit of reducing T2DM related comorbidities and mortality as well as limiting the use of glucose lowering medication [55].

An exercise program which improves any of the MetS markers may result in a reclassification of patients who may no longer meet MetS criteria. The combined effect in improvement of MetS markers following 20 weeks of supervised exercise in healthy adults, led to reclassification of about 30.5% of participants as no longer having MetS [58]. Therefore, a marginal reduction in WC, as shown in this meta-analysis, has the potential to improve patient health outcomes including reducing the risk of cardiovascular disease.

Our analysis found that following resistance training, MetS markers in adults with T2DM remain unaltered. However, there was a trend towards improvement. A meta-analysis on healthy adults diagnosed with MetS without diabetes showed that glucose-insulin mechanism did not improve following supervised exercise [53]. The authors postulated that one reason may be due to the study recruiting healthy participants without diabetes. However, our findings in this meta-analysis involving diabetes patients did not reflect the conclusion that glucose-insulin mechanism may improve in people with T2DM after undergoing resistance training.

Similarly, our findings indicate a change in HDL following aerobic exercise (2.0 mg/dL increase) and resistance exercise (0.25 mg/dL increase). However, those changes were statistically non-significant. In a recent meta-analysis on the effect of aerobic exercise on MetS markers in healthy older adults, there was significant improvement in HDL and triglycerides [59]. This contrasts with our findings in which there was no statistical difference in MetS markers following aerobics and resistance exercise in adults with T2DM. A plausible reason for this difference could be the participant characteristics (healthy versus patients with diabetes), as cited in the study by Pattyn et al. [53].

There were several limitations to this meta-analysis. Firstly, allocation concealment and blinding participants and therapist is difficult to achieve [60], particularly in exercise intervention studies. Therefore, there is the likelihood of bias in such studies. A study that examined the quality of evidence for RCTs involving PA interventions found that less than half of RCTs reported the intention-to-treat principle, achieved allocation concealment, and had blinded assessors, therapists and participants [60]. In our current study, only one study achieved participant blinding and four achieved therapist blinding. Half of the studies (n = 13) in the current study reported intention-to-treat analysis. Almost half of the studies (n = 12) also achieved assessor blinding. Strategies in achieving reduced research bias in those areas should be explored in further studies.

Another limitation is that several definitions of MetS exist including definitions by World Health Organisation, International Diabetes Federation, and NCEP-ATP III, with different cardiovascular risk markers. Not all the definitions were considered in this study. Only studies which met the ATP III criteria [10] for metabolic syndrome were included. Again, sub-analysis on the impact of PA relative to intensity and duration, and combined exercise, was not done due to limited studies. More research is needed to investigate the impact of combined exercise on MetS markers in people with T2DM.

## 5. Conclusions

Overall, our results indicate that aerobic exercise can improve WC in people with T2DM. However, both aerobic and resistance exercise produced no significant difference in the remaining MetS markers, despite evidence of a trend towards improvement for some markers. Studies regarding the effect of combined (aerobics plus resistance) exercise on MetS markers is limited, therefore further larger, multicentre RCTs are warranted.

## Figures and Tables

**Figure 1 sports-11-00101-f001:**
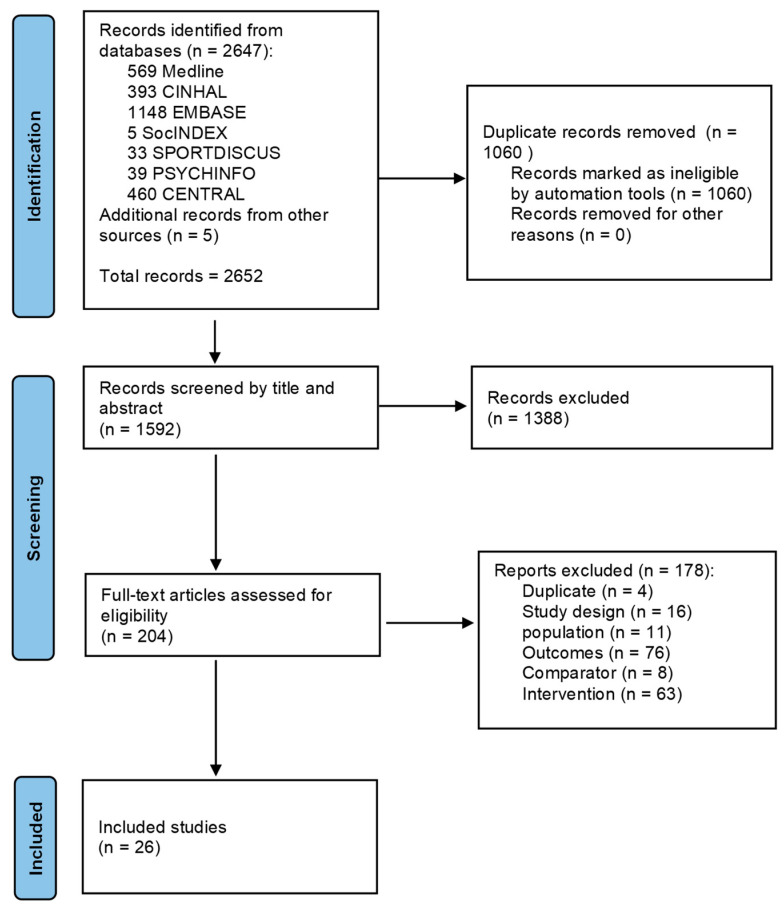
PRISMA flow chart.

**Figure 2 sports-11-00101-f002:**
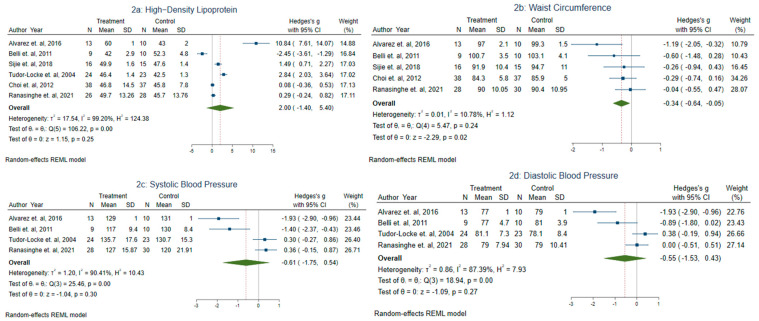
Effect of aerobic exercise on MetS markers in T2DM. Data are reported as Hedge’s G (effect size) and 95% confidence interval (CI). The diamond at the bottom presents the overall effect. The plotted squares denote effect sizes, and the whiskers denote their 95% CIs [2,5,7,18,22,23].

**Table 1 sports-11-00101-t001:** Risk of bias assessment of studies.

Study Author, Year	Specified Eligibility Criteria	Random Allocation	Allocation Concealed	Groups Similar	Blinding of All Subjects	Therapist Blinding	Assessor Blinding	Point Estimate Reported	<15% Dropout Rate	Intention-to-Treat Analysis	Group Difference Reported	Total PEDro Score
Agurs-Collins et al., 1997 [1]	🗸	🗸	-	-	-	-	-	🗸	🗸	-	🗸	5
Alvarez et al., 2016 [2]	🗸	🗸	🗸	🗸	-	-	🗸	🗸	-	-	🗸	7
Balducci et al., 2010 [3]	🗸	🗸	🗸	🗸	-	-	🗸	🗸	🗸	-	🗸	8
Bassi et al., 2016 [4]	🗸	🗸	🗸	🗸	-	-	🗸	🗸	🗸	-	🗸	8
Belli et al., 2011 [5]	🗸	🗸	🗸	🗸	-	-	-	🗸	-	-	🗸	6
Castaneda et al., 2002 [6]	🗸	🗸	-	-	-	-	🗸	🗸	🗸	🗸	🗸	7
Choi et al., 2012 [7]	🗸	🗸	-	🗸	-	-	-	🗸	🗸	-	🗸	6
Church et al., 2010 [8]	🗸	🗸	-	🗸	🗸	🗸	🗸	🗸	🗸	🗸	🗸	9
Conners et al., 2017 [9]	🗸	🗸	🗸	-	-	-	-	🗸	🗸	🗸	🗸	7
Emerenziani et al., 2015 [10]	🗸	🗸	-	🗸	-	-	-	🗸	🗸	-	🗸	6
Gordon et al., 2008 [11]	🗸	🗸	-	-	-	-	🗸	🗸	🗸	🗸	🗸	7
Hangping et al., 2019 [12]	🗸	🗸	🗸	🗸	-	-	🗸	🗸	🗸	-	🗸	8
Hsieh et al., 2018 [13]	🗸	🗸	🗸	🗸	-	🗸	🗸	🗸	🗸	🗸	🗸	9
Huimin et al., 2014 [14]	🗸	🗸	-	🗸	-	-	-	🗸	🗸	🗸	🗸	7
Lam et al., 2008 [15]	🗸	🗸	🗸	-	-	🗸	🗸	🗸	-	-	🗸	7
Lambers et al., 2008 [16]	🗸	🗸	🗸	🗸	-	-	-	🗸	🗸	🗸	🗸	8
Plotnikoff et al., 2010 [17]	🗸	🗸	🗸	🗸	-	-	🗸	🗸	🗸	🗸	🗸	8
Ranasinghe et al., 2021 [18]	🗸	🗸	🗸	-	-	🗸	🗸	🗸	-	🗸	🗸	8
Shantakumari et al., 2013 [19]	🗸	🗸	-	-	-	-	-	🗸	🗸	🗸	🗸	6
Shenoy et al., 2009 [20]	🗸	🗸	🗸	-	-	-	🗸	🗸	🗸	-	🗸	7
Sigal et al., 2007 [21]	🗸	🗸	🗸	🗸	-	-	-	🗸	🗸	🗸	🗸	8
Sijie et al., 2018 [22]	🗸	🗸	🗸	🗸	-	-	-	🗸	🗸	-	🗸	7
Tudor-Locke et al., 2004 [23]	🗸	🗸	-	🗸	-	-	-	🗸	-	-	🗸	5
Vancea et al., 2009 [24]	🗸	🗸	-	🗸	-	-	-	🗸	🗸	🗸	🗸	7
Wang et al., 2019 [25]	🗸	🗸	-	-	-	-	-	🗸	🗸	🗸	🗸	6
Yavari et al., 2012 [26]	🗸	🗸	-	-	-	-	-	🗸	-	-	🗸	4

Note: (🗸) = Yes, (-) = no; Total PEDro scale ranges from 0 to 11, where scores closer to 11 indicates low risk of study bias.

## Data Availability

Data are contained within the article or Appendix A.

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
