# Peer review of "Effect of Physical Activity on Metabolic Syndrome Markers in Adults with Type 2 Diabetes: A Systematic Review and Meta-Analysis"

_sports, 2023, doi:10.3390/sports11050101_

Round 1

Reviewer 1 Report

This systematic review and meta-analysis reports the findings of previous literature investigating the role of physical activity to modulate MetS markers in people with T2DM. The topic is of interest and since the plenty of original works, good reviews are well-received.

The approach to the meta-analysis looks appropriate, the analysis of bias is well-performed and the statistical methods correct. The way the work is presented is clear and help the reader to have a direct understanding of the state-of-the-art on this topic. I have no further comments and congratulate with the authors.

Author Response

Notes attached

Reviewer 2 Report

Dear Authors,

I have read your manuscript with interest, however I have few minor comments/suggestins:

1. Maybe you can propose some other keywords that these which are in the title?

2. Aim of the study should not be in the form of question.

3. If it is a systematic review, some additiolnal table with more qualitative information about the RCTs studies from the articles taken into the analysis would be valuable. E.g. the reder do not need to look into the referenced article directly but can see in the table what was the no. participants or the main result etc.

4. I would remove subheading 6.

Author Response

Notes attached

Reviewer 3 Report

Amin_physical activity on metabolic syndrome markers_sports_2023. Reviewer report

Thank you for the opportunity to review this manuscript. It presents a very wide revision of the literature on a troubling topic. However, I have some comments and suggestions presented below.

General comments. 

Please, include the developers and/or web address of the applications used to perform the manuscript. Ex. Covidence.

Abstract

Line 32. Please change I2 to IIndex. And also correct this issue in the rest of the manuscript. 

Line 35. Please change WC to waist circumference. 

1. Introduction

Line 79. Please change Sugar to glucose. 

2. Materials & Methods

Line 92. Please change FBS to FBG. And also correct this issue in the rest of the manuscript. 

Line 115: Supplementary file 1. Legends: Please change: + to *

Lines 139 and 140. Please change Data was to Data were. I believe that it is more correct.  And also correct this issue in the rest of the manuscript. 

Line 156: supplementary file 2: Estimated θ. Please explain to what refers θ in a legend under the Forest plot. I think it is the odds of a positive outcome under the treatment. 

Line 157. Please change supplementary file 2 to supplementary file 2 and 3. 

3. Results

Line 177. Please change supplementary file 3 to supplementary file 6 or to supplementary file 3-6.

Line 229. Please change Effect of aerobic training to effect of resistance training. 

I think it is better to include Figure 2a-d.: Effect of aerobic exercise on MetS markers in T2DM in the supplementary file 4.

Please refers to supplementary files 4 and 5, in the main manuscript text when necessary (Subheadings 3.3.1. and 3.3.2.)

Please consult your statistician. The values you present as effect size in the text of the paragraphs Subheadings 3.3.1. and 3.3.2. are the p values of the effect size, and for this reason it is correct to say that they are or are not significant, but they are not absolute values of effect size. I suppose the second value of θ in the plots are the effect size values. 

4. Discussion

Line 250. Please change small effect size to small difference of means. 

Line 268. Please change MetS markers to glucose metabolism markers. 

Line 283. Please change “the study by” to “the study by Pattyn et. al.”

Author Response

Notes attached

Round 2

Reviewer 2 Report

No comments.

Reviewer 3 Report

I believe that the manuscript has been adequately improved and is ready for publication.